# Prediction of the Levodopa Challenge Test in Parkinson’s Disease Using Data from a Wrist-Worn Sensor

**DOI:** 10.3390/s19235153

**Published:** 2019-11-25

**Authors:** Hamid Khodakarami, Lucia Ricciardi, Maria Fiorella Contarino, Rajesh Pahwa, Kelly E. Lyons, Victor J. Geraedts, Francesca Morgante, Alison Leake, Dominic Paviour, Andrea De Angelis, Malcolm Horne

**Affiliations:** 1Global Kinetics Pty Ltd., 31 Queens St., Melbourne 3000, Australia; hamid.khodakarami@globalkineticscorp.com; 2Neurosciences Research Centre, Molecular and Clinical Sciences Research Institute, St George’s University of London, Crammer Terrace, London SW18 0RE, UK; lucia.ricciardi2@gmail.com (L.R.); fmorgant@sgul.ac.uk (F.M.); a.leake@nhs.net (A.L.); dpaviour@sgul.ac.uk (D.P.); andreadeangelis.md@gmail.com (A.D.A.); 3Department of Neurology, Leiden University Medical Center, K5-Q103 Albinusdreef 2, 2333 ZA Leiden, The Netherlands; M.F.Contarino@lumc.nl (M.F.C.); V.j.geraedts@lumc.nl (V.J.G.); 4Department of Neurology, Haga Teaching Hospital, Els Borst-Eilersplein 275, 2545 AA The Hague, The Netherlands; 5Parkinson’s Disease and Movement Disorder Center, University of Kansas Medical Center, 3599 Rainbow Blvd, MS 3042, Kansas City, KS 66160, USA; RPAHWA@kumc.edu (R.P.); kelly.lyons@att.net (K.E.L.); 6Department of Clinical and Experimental Medicine, University of Messina, 98122 Messina, Italy; 7Florey Institute of Neuroscience and Mental Health, University of Melbourne, Melbourne, VIC 3010, Australia; 8St Vincent’s Hospital, Fitzroy 3065, Australia

**Keywords:** levodopa challenge test, Parkinson’s Disease, levodopa response, wearable devices, ambulatory systems, machine learning

## Abstract

The response to levodopa (LR) is important for managing Parkinson’s Disease and is measured with clinical scales prior to (OFF) and after (ON) levodopa. The aim of this study was to ascertain whether an ambulatory wearable device could predict the LR from the response to the first morning dose. The ON and OFF scores were sorted into six categories of severity so that separating Parkinson’s Kinetigraph (PKG) features corresponding to the ON and OFF scores became a multi-class classification problem according to whether they fell below or above the threshold for each class. Candidate features were extracted from the PKG data and matched to the class labels. Several linear and non-linear candidate statistical models were examined and compared to classify the six categories of severity. The resulting model predicted a clinically significant LR with an area under the receiver operator curve of 0.92. This study shows that ambulatory data could be used to identify a clinically significant response to levodopa. This study has also identified practical steps that would enhance the reliability of this test in future studies.

## 1. Introduction

Parkinson’s Disease (PD) is a progressive neurodegenerative disorder that affects the frontal lobe, the brainstem and the autonomic nervous system. Impaired dopamine transmission is a defining feature of PD and can be treated by pharmacological therapies such as levodopa. Evidence of responsiveness to levodopa provides support for the clinical diagnosis of PD and for assessing the suitability for device assisted therapies such as deep brain stimulation (DBS) and delivery of apomorphine or levodopa by pump.

Responsiveness to levodopa is often assessed with the levodopa challenge test (LDCT) [1], following a protocol described in 1999 [2], in which the test is performed in the morning, having ceased levodopa for 12 h and dopamine receptor 2 agonists (D2 agonists) for 24 h. The improvement from OFF to ON is known as the levodopa response (LR). Bradykinesia is a central clinical feature of diminished dopamine transmission and this is assessed, along with tremor and limb rigidity, using a clinical scale known as the Unified Parkinson’s Disease Rating Scale Part III (UPDRS III).

Although LDCT has been used for both research and clinical purposes for more than 30 years [3], there is remarkable variation in the manner in which it is deployed and interpreted [4]. Despite these problems, the LDCT has become accepted as a benchmark measure of responsiveness to levodopa. The LDCT is often done as an ambulatory test, requiring an early morning trip to the hospital in the “OFF” state and in some cases an overnight stay. It is thus inconvenient, uncomfortable [4], time consuming and with costs for the clinical centers and not without complications [5,6]. An alternative means of assessing responsiveness to levodopa that eliminated these problems would therefore be welcome.

The recent development of objective measures of the motor features of PD raises the possibility of assessing LR in the home were reviewed by Maetzler et al. [7,8] and Espay et al. [9]. However, these reviews indicate there are only one or two of these systems that actually measure bradykinesia while subjects are engaged in routine daily activities. One of these ambulatory measuring systems is the Parkinson’s Kinetigraph (PKG), which scores the motor features of PD [10,11], without the need for the subject to engage in specific test activities. The data available from a PKG recording includes the response to the first dose of levodopa each day for 6 consecutive days. The PKG is obtained by a Person With Parkinson’s Disease (PwP) wearing a logger for 6 days while taking their usual medications and attending to their usual activities in the home. Our question in this study was whether the LR measured during the LDCT (LR_UPDRS_) could be predicted from the change in motor function measured by the PKG following the first levodopa dose in the morning (LR_PKG_).

There are challenges to using this standard PKG to assess the LR. In a standard 6-day PKG, the first dose of levodopa is usually not supramaximal and D2 agonists may be taken. Furthermore, some PwP choose to continue to rest for some time after their first dose to allow it to take effect and this immobility may obscure the presence of bradykinesia. On the other hand, 6 days of data are available, and this may address some of the natural variation in response to levodopa that will be overlooked by a single dose administration.

In this paper, we describe how data from the UPDRS III in the “OFF” and “ON” state (UPDRSIII_ON_ and UPDRSIII_OFF_) during an LDCT and features from the PKG data were used to build a model of motor function severity levels (MFSL) using Logistic Regression. This model was then used to predict abs∆ and %∆ from the LDCT calculated using the PKG data.

## 2. Materials and Methods

The data used in this study was collected from 199 people with PD (PwP) who attended one of the three clinics (St George’s, Leiden University Medical Center/Haga Teaching Hospital or the University of Kansas Medical Center) for an LDCT as part of routine clinical care. In the case of St George’s and Leiden/The Hague, the data was collected and collated as part of other studies approved by institutional ethics with approval to use the deidentified data in related studies. In Kansas, the data was collected and collated as part of routine clinical care with institutional approval to use the deidentified data. As well, data from 191 people without PD and aged >60 was used under approval provided by St Vincent’s Hospital Melbourne Human Research & Ethics Committee. PKGs were also available from 24 subjects who underwent deep brain stimulation (DBS). PKGs were performed prior to DBS and months after DBS. This study was carried out in accordance with the guidelines issued by the National Health and Medical Research Council of Australia for Ethical Conduct in Human Research (2007, and updated May 2015). Clinical characteristics of participating subjects and inclusion criteria into the analyses are discussed below.

### 2.1. Patient Cohort

Data from the 199 PwP who underwent a LDCT as part of routine clinical care at these 3 clinics were reviewed. Subjects were excluded if; (a) a PKG logger had not been worn at or near the time of the LDCT (median 6 days and 75th percentile 22 days); (b) it was documented that PD medications were consumed prior to the first PKG medication reminder; (c) PD medications were used at the time of the first dose whose pharmacological profile would mask or confound the usual response to levodopa (apomorphine, levodopa in intestinal gel formulation, levodopa in extended release capsules (Rytary) but not D2 agonists or entacapone); (d) there was no reminder set on the PKG at or around the time of awakening; (e) the first dose occurred before 5:00 am. Of the original 199, 48 were excluded for one or more of the above reasons. The clinical features of the remaining 151 PwP are shown in Table 1.

Controls were 191 subjects aged 60 years and over, without a known neurological disorder and who had worn a PKG logger. Subjects whose bradykinesia scores (as measured by the PKG) were >90th percentile for controls were removed. Thus, there were 174 control subjects. The reason control subjects were included was to provide a large cohort of people who would not be bradykinetic in the early morning and whose bradykinesia score would not change after a nominal “dose time” (DT). DT was 07:00 for building the model: this was chosen to ensure that levels of sleep and inactivity would be similar to the first dose in the morning of PwP. For estimating the delta of the LDCT, DT was 10:00 and chosen explicitly to avoid sleep and inactivity. The UPDRS III was deemed to be “0” at DT.

### 2.2. The PKG System

The PKG system consists of a wrist-worn data logger (the PKG logger), a series of algorithms that produce data points every two minutes and a series of graphs and scores that synthesize this data into a clinically useful format known as the PKG. The PKG system [10] was the first system to receive clearance from the United States Food and Drug Administration to measure bradykinesia, and also has indications for measuring dyskinesia, tremor, and sleep.

The logger is a smart watch that is worn on the most affected wrist (see http://www.globalkineticscorporation.com.au/ for details). It weighs 35 g and contains a rechargeable battery and a 3-axis accelerometer set to record 11-bit digital measurement of acceleration with a range of ± 4 g and a sampling rate of 50 samples per second using a digital micro-controller and data storage on flash memory. The device is water resistant. The logger has been designed and approved for easy cleaning and reuse and after 6 days of recording, data are uploaded to the cloud for application of the algorithms. The algorithms were built using an expert system approach to model neurologists’ recognition of bradykinesia and dyskinesia on accelerometry data. Inputs to the expert system included Mean Spectral Power (MSP) within bands of acceleration between 0.2 and 4 Hz, peak acceleration, and the amount of time within these epochs that there was no movement. These inputs were weighted to model neurologists’ rating of bradykinesia and dyskinesia and to produce a bradykinesia score (BKS) and dyskinesia score (DKS) every 2 min.

The PKG is the graphical representation of the output of the algorithms from data collected every 2 min over an extended period, typically 6 days. The PKG plotted the BKS and DKS against the time of day. The time that medications were due and consumed were also shown, making it possible to assess whether there were dose related variations in BKS or DKS and how the median value at any time of day compared with a normal subject. Other scores are also provided, including compliance with the reminders, tremor scores [13], and times when the logger was not worn. The system also provides scores of day sleep and inactivity [14]. The numerical output relevant to this study is summarized further in the Glossary that follows. The reader is referred to the company’s web site for further details (http://www.globalkineticscorporation.com.au/) and to other publications [10,11,13,14,15,16,17,18,19,20]. Specific scores used in this paper are described in further details below.

### 2.3. Glossary of PKG Terms

•Bradykinesia Score (BKS). The score for bradykinesia produced by the PKG algorithm for a 2 min epoch. These inputs were weighted to model neurologists’ rating of bradykinesia. BKS are produced every 2 min while the logger is being worn.•Epoch. In this paper, “epoch” will refer to a 2-min period in which PKG scores are calculated.•Tremor Amplitude (PA): This is the 75th percentile of the magnitude of the tremulous peak identified from the spectrogram from each epoch. Epochs without tremor are scored zero.•OFF Wrist: The logger has a capacitive sensor to detect whether it was being worn. An epoch marked as “OFF wrist” was unavailable for analysis.•Dose reminder: The logger can be programmed to vibrate for 10 s at the time when levodopa should be taken as a reminder to the wearer to take medications.

Dose acknowledgement. Following a reminder, the smart screen on the logger can be “swiped” by moving the finger slowly across the screen to indicate when the medication was consumed.

There are some terms that specifically apply to the first dose of the day (Figure 1) and relate to the presumption in this study, that the level of motor function at the time of this first dose would be similar to the UPDRSIII_OFF_ and the response to this first dose would be similar to the UPDRSIII_ON_.

•Dose Time (DT). This is the 5 epochs (10 min), centered on the acknowledgement to the first reminder of the day and shown as a red diamond in Figure 1. It may be a few minutes after the reminder was delivered (as in Figure 1).•Effect Time (ET): This is the time when levodopa had its peak effect. It is calculated for the PKG as the peak of Smoothed Weekly BKS Time series from 46 min to 90 min after the acknowledgement of the dose reminder.•Active epochs: Epochs whose linearly weighted moving median of BKS ≤ 0.•Inactive epochs: Epochs whose linearly weighted moving median of BKS > 40 were attributed to inactivity or sleep and excluded [14].•Reminder Time: The PKG logger is programmed to vibrate at specific times to remind subjects to take their medications. Subjects can acknowledge when they actually consume the medications by “swiping” the smart screen on the watch. The first acknowledgement of the reminder is shown as a red diamond in Figure 1.

The 5 epochs (10 min) centered on DT and ET were used to calculate further features of motor function from the PKG (see Section 2.7. Feature Engineering and Selection for more detailed discussion). These features are:•Weekly Aggregate Mean: Epochs from each Dose time (or Effect time) are aggregated: 5 from each dose and potentially, 6 dose times from the 6 days (week) of recording. Thus potentially, 30 epochs in total are aggregated and these are averaged to produce the weekly aggregate mean.•Weekly Aggregate Standard Deviation: Epochs from each Dose time (or Effect time) are aggregated: 5 from each dose and potential 6 dose times from the 6 days (week) of recording. Thus potentially, 30 epochs in total are aggregated and the standard deviation of the aggregate is calculated to produce the weekly aggregate standard deviation.•Thirty-minute window moving percentiles of BKS. The 10th, 25th, 50th, 75th and 90th percentile of the BKS value of the 15 epochs in the window around the first epoch within the DT (or ET) was estimated. The window was then slid for the five epochs within the DT (or ET) to calculate the 30-min moving percentiles of BKS. The 30-min moving percentiles approximate the shape of the distribution of a window of 30 min around each epoch. They will be referred to as, for instance, BKS_M75P for the 30 min window moving 75th percentile of BKS.•Thirty-minute window weighted moving percentiles of BKS. Similarly, a 30-min weighted moving percentile of the BKS were also calculated. The weighting was linear, being maximal at the center of the window and declining linearly and symmetrically on each side to zero and provides some indication of the influence of shorter window sizes. They will be referred to as, for instance, BKS_WM75P for the 30 min window weighted moving 75th percentile of BKS.•Thirty-minute window moving 50th percentile of Tremor Amplitude. This is calculated in a similar manner to the 30-min moving percentiles of BKS and will be referred to as TA_M50P.•Thirty-minute window weighted moving 50th percentile of Tremor Amplitude. This was calculated in a similar manner to the 30-min window weighted moving percentiles of BKS and will be referred to as TA_WM55P.•Logarithm of 1 + 30 min window weighted moving 50th percentile of Tremor Amplitude: While the BKS is linearly correlated with UPDRS III [10] and scales logarithmically with respect to the acceleration of movement, the Tremor Score requires translating into the same scaling by defining Log (1 + Tremor Score). The “1” is added to obtain a finite non-negative feature. For example, the Log10(1 + 30 min window weighted moving 50th percentile of Tremor Amplitude) will be referred to as TA_WM50P_Log.

### 2.4. The Unified Parkinson’s Disease Rating Scale Part III

The UPDRS Part III is a formalized version of the clinical examination with scores (usually 0–4) for various severities of motor function. The score of each of the items is summed to give a total score. There are two versions of the UPDRS III: the older is known simply as the UPDRS III and the newer version, which is owned by The Movement Disorder Society is termed the MDS-UPDRS III. Six items were added to the scale in the MDS-UPDRS III, so that the maximum possible total score increased from 108 to 132 points. Unfortunately, both versions are frequently used in practice. Formulae for allowing the older and newer total scores to be used have been proposed [21] but we have used a method that requires the addition of 7 points to the older scale to produce a robust correction [22].

In this study, no subjects had missing values in the UPDRS III. Two clinics used the older UPDRS III scale and one used the MDS-UPDRS III scale. As only the total scores were used, the older scales were adjusted by the addition of 7 points and UDPRS III is used from here to refer to the adjusted score. It is important to note that in the UPDRS III, scores for tremor and bradykinesia contribute ~30% and ~40% (respectively) whereas the BKS of the PKG only relates to bradykinesia with a separate score for tremor.

### 2.5. The LDCT

As described earlier, the LDCT is performed by administering a dose of levodopa in the “OFF” state which is then compared with the “ON” state using the UPDRS III. As already noted, there is considerable variation in the performance of this test, which was also apparent in the procedures used by the three clinics participating in this study.

•OFF state assessment. All clinics assessed the “OFF” state at least 12 h after the PwP’s last PD medication. No clinic ceased D2 agonist for more than 12 h.•Loading dose. One clinic gave a supramaximal dose of 50% more than their usual first dose of levodopa. One clinic gave 20% more than the usual morning dose in dispersible form. The third reviewed the “ON” state on their usual dose•ON state assessment. Two clinics assessed the “ON” state following the levodopa dose: one from 45 min on and the other at about 50–60 min. The third clinic made the assessment on another day to when the “OFF” state was assessed.

### 2.6. The PKG Representation of the LDCT

The PKG records the first dose on six consecutive mornings and the presumption in this study was that the level of motor function at the time of this first dose would be similar to the UPDRSIII_OFF_. Similarly, the response to this first dose should be similar to the UPDRSIII_ON_. As “ON” and “OFF” have specific clinical meanings, in this study the terminology of motor function at DT (or ET) is used to signify motor function measured by the PKG (see Figure 1). In modelling the LDCT using the PKG in this way we recognize the following issues:•The first levodopa dose of the morning may be substantially less than the dose of levodopa used in the LDCT. In fact, the average levodopa in the first dose was 164 mg (±79 SD) and only 25% of PwP took 200 mg or more at the first morning dose (Figure 2).•D2 agonists were taken by 68% of PwP, in which case it constituted a median of 15% (8%–30% Inter-quartile range) of the total levodopa equivalent dose. The size of the levodopa response measured using the UPDRS III, was not smaller in those on D2 agonists (even in those where it constituted more than 25% of the levodopa equivalent dose (Figure 2 and Table 1).•Some subjects may have remained in bed or resting during the dose time, so that all epochs that were inactive epochs (see definition above) or OFF wrist were excluded, and Table 1 shows the number of cases whose dose time was available for modeling UPDRSIII_OFF_.•Similar exclusions occurred on examination of the ET, and Table 2 shows the number of cases whose dose time was available for modeling UPDRSIII_ON_.•In some subjects, especially those with impaired gastric emptying, there may be variability in the time to peak effect and in the amplitude of the peak effect. Thus, a single day captured by the LDCT may not represent the summary of the 6 days from the PKG.

These factors are all addressed further in either the model or the analyses of the LDCT in the results.

### 2.7. Model Design for Response Estimation

The first step in designing the computational model that estimates the response to a dose of levodopa was to divide the UPDRS III into a range of motor function severity levels (MFSL_UPDRS_). This was to allow severity of motor function to be categorically classified instead of requiring regressions. This allows a more robust employment of the dataset with the non-identically distributed noise and variations in UPDRS III scoring, which introduces complexity for regression models. MFSL_UPDRS_ “0” was set as a UPDRS III score <10 and levels above that were separated by increments of 12.5 UPDRS III units, with all scores ≥60 in level 5 (See Table 3). Unfortunately, the literature is relatively silent on the UPDRS III of normal subjects, and while a score of UPDRS III < 10 for MFSL_UPDRS_ “0” is somewhat arbitrary, it is designed to be conservatively low, based on the range of scores in newly diagnosed subjects [23] and on the clinical experience of the authors. The size of the increments in UPDRS III between higher MFSL_UPDR_ relate to the upper limits of inter/intra rater variability [21,24] and clinical important differences [25], although we accept that there is little literature to guide this choice. Table 3 shows the number of UPDRS_OFF_ and UPDRS_ON_ (or support sets) in each MFSL_UPDRS_. The various PKG features (Table 4 and glossary (Section 2.3) used to measure motor function were assigned to one of the 6 MFSL class labels (Table 3) using the corresponding UPDRS III score.

We decompose this multi-class classification problem into 5 binary classification problems, where samples are separated according to whether they fell below or above the threshold for each class. For instance, Classifier 3 will be a binary classifier separating UPDRS III < 35 and UPDRS III ≥ 35. The next section explains the design of these classifiers.

### 2.8. Feature Engineering and Selection

A set of candidate features (enumerated in Table 4 and in the glossary, Section 2.3) for performing the classification task were extracted from the PKG data and statistically matched to the class labels in Table 3. When dealing with a relatively small dataset, having many structurally dependent features can degrade the performance of the statistical model and risks overfitting due to noise induced variance. What follows is a reduction in the size of feature space by elimination and selection to enhance the model generalizability and interpretability in a supervised manner. The first phase of feature selection was to identify features of same nature and select the most relevant. For example, the two features BKS_M50P and BKS_WM50P in Table 3 are of the same nature in that they are both moving medians of BK Score and would thus contain similar information.

The mutual information (MI) test was performed to assess the relevance of these features to the target classes. This approach is neutral with respect to models and identifies any statistical relationship, linear or nonlinear, between the features and the class labels [26]. Table 4 shows the MI of each feature with the 6 target classes, estimated using 6 nearest neighbors for these continuously valued features, with the 6 class labels.

Based on the above discussion, the feature set was refined to BKS_M10P, BKS_M25P, BKS_M50P, BKS_WM75P, BKS_M90P and TA_WM50P_Log. Although these scores each have considerable relevance, some carry redundant information with respect to one or a combination of the others. Joint Mutual Information Maximization (JMIM) [27] was used to maximize relevancy while minimizing redundancy. This method first picks the feature with maximal MI with target classes and adds it to the set of selected features. It then iteratively adds the feature for which the minimum joint mutual information together with any of the already selected features is maximum among other candidates for selection. This heuristically ensures that a newly selected feature has greater relevancy and less redundancy. Table 5 shows the ranking of iterative selection of the refined feature set applying JMIM. Also shown are the mutual information of Weekly Aggregate Mean of each feature, i.e., the average of values of the features over a 10 min interval of entire week, with target classes. This shows the relevance of a feature when aggregated at DT and ET. Evidently, BKS_M90P and BKS_WM75P were relatively less relevant. Furthermore, BKS_M50P was relatively redundant and was also less relevant after weekly aggregation. Although there was the high level of collinearity expected between BKS_M10P BKS_M25P, both features were deployed as they represent different nonparametric measures of the temporal distribution of BKS. In summary, we selected the set of features namely BKS_M10P, BKS_M25P and TA_WM50P.

### 2.9. Model Selection, Training and Validation

Several discriminative statistical models were examined for the purpose of designing the five binary classifications algorithms. Although the selected features are monotonically and linearly correlated with the UPDRS III, linear and nonlinear models were considered: namely Logistic Regression [28], Support Vector Classifier [29,30] with Radial Basis Function (RBF) kernel and Gradient Boosting Decision Trees [31]. These discriminative models represent categories of linear, kernel nonlinear and ensemble of arbitrarily nonlinear approaches which have proved to be effective in a variety of problem types. The three features (BKS_M10P, BKS_M25P and TA_WM50P_Log) used are defined above.

When comparing the performance of the learning models described above, it was noted that Logistic Regression assumes that the observations in the dataset are independent and that multicollinearity is not present. We previously noted the structural collinearity between BKS_M10P and BKS_M25P. To investigate the effect of the violation of this assumption on Logistic Regression, we introduce a fourth model, where unsupervised reduction of the dimension of these two features into one is achieved using the first component of Principal Component Analyses (PCA) of BKS_M10P and BKS_M25P together with TA_WM50P_Log, followed by a Logistic Regression classifier. Furthermore, as the employed features are moving statistics, neighboring epochs (observations of the same subject) are not independent when 5 epochs in a row are sampled. Two strategies were used to address assumptions regarding independence of observations: (i) using other candidate models whose requirements around this assumption are relaxed (SVM and Gradient Boosting Trees), (ii) assessing the performance of the models both in terms of model selection and out of sample predictions on Weekly Aggregate predictions, where two observations from one subject (weekly aggregate of dose time and effect time) are spaced more than 30 min apart and are thus reasonably independent. Therefore, the use of Logistic Regression as one of the candidate classifiers is possible and the performance comparisons would be meaningful.

A train set and a test set were defined for each of the classification problems (i.e., into one of the levels defined in Table 3). For each binary classification problem, we allocate 30% samples from each class around Dose and Effect Time of the entire set to test set and use the remaining in train set. As binary classes for each classifier are different, train and test set will randomly vary across them. We ensure that if samples from a subject are used in test set, samples from that subject are not used in train set. These imply that we retain the original distribution balance in test set and ensure the complete separation of train and test sets for each classification problem.

We report two performance measures: (i) area under curve of Receiver Operating Characteristics (ROC AUC) which provides an average Sensitivity at different Specificities; (ii) area under curve of Precision (ratio of true positives to all positive declarations including false positives) vs. Recall (=Sensitivity) (PR AUC) which provides an average Precision at different Recalls averaged over the two classes. The performance metric (ii) is reported to account for the class imbalance that varies over the five Classifiers.

To tune the critical hyperparameters of each model, and assess their generalisability, a 10-fold hold-out cross validation (CV) on Train set was performed (Table 6). First it is evident that reducing the dimensionality of the structurally collinear features with PCA did not improve the performance of the Logistic Regression. This will support the employment of BKS_M10P and BKS_M25P together as separate features in Logistic Regression model that allows to capture the distribution of BK score reflected by these two features. Second, the RBF kernel SVC and Gradient Boosting Decision Trees model do not perform as well as Logistic Regression on Classifiers 4 and 5 while not offering any advantage at Classifiers in lower levels. Table 6 shows the performance of these models of various complexities on train set. Although Gradient Boosting Decision Trees model offers a lower bias than Logistic Regression for all classifiers, the cross-validation performance shows that the variance was not convincingly resolvable through parameter tuning.

The Logistic Regression model performs as well, if not better than the other models and is simpler in that it enhances the likelihood of generalizability and interpretability of the algorithm. Thus, the Logistic Regression model was further analyzed for validation and bias-variance assessment using the five classifier models and Logistic Regression was used in the following sections of this study.

The generalizability of the classifier models was examined by applying them to an unseen test set. Train and test sets for each classifier model were selected according to the description in the previous section. We decomposed this multi-class classification into 5 binary classification problems, where samples were separated according to whether they fell below or above the threshold for each class. Table 7 shows the performance of each classifier model on train and test sets.

## 3. Results

The aim of this study was to use the model described above to compare the response to levodopa (LR) measured during the LDCT with the change in motor function following the first morning dose measured by the PKG over 6 days. To remind the reader, the LR in the LDCT (LR_UPDRS_) is calculated as the change in the UPDRS III scores (abs∆_UPDRS_) from the OFF state to the ON state, which was 22 (±11 SD) UPDRS III units for the whole cohort (see Table 1 for each clinic). The LR_UPDRS_ is also commonly expressed clinically as %∆_UPDRS_ (abs∆_UPDRS_/UPDRSIII_OFF_ × 100), or the percent improvement, which was 47 (±18 SD) for the whole cohort (see Table 1 for each clinic). Figure 3 shows the relationship between abs∆_UPDRS_ and %∆_UPDRS_. We compare the model with the abs∆_UPDRS_ first and then with %∆_UPDRS_ and the justification for considering the abs∆ is fully reviewed in the Discussion.

For the PKG, the Logistic Regression model designed in Section 2.9 was used to obtain a binary prediction for each epoch in the DT and ET as to whether it was below or above each MFSL At DT, the estimated motor function score for all available epochs were averaged to produce the PKG estimate of the motor function score at DT (MFSL_DT_), and similarly at ET(MFSL_ET_). These were used to produce the model’s estimate of the LR: abs∆_PKG_ (MFSL_DT_-MFSL_ET_) and %∆_PKG_ (abs∆_PKG_/MFSL_DT_ × 100).

Controls were used to augment the number of cases whose LR could be considered insignificant (Support Class 0 for ROC) and ensure class balance. For controls, DT was set at 10:00 a.m. and ET was at the peak of BKS smoothed weekly summary from 45 min to 90 min. UPDRS III was set at “0”. The PwP and Controls included in the assessment of the response to the first morning dose of levodopa in the PKG as a model of the LDCT are shown in Table 2.

### 3.1. Performance of the PKG Predictions of LR Compared to the UPDRS III Measurement of LR

The performance of the abs∆_PKG_ as a prediction of abs∆_UPDRS_ was estimated in terms of the two metrics of Receiver Operator Characteristic and Precision-Recall Curve (See Section 2.8 for definition) (Table 8). When all subjects (column headed “All” in Table 8 and Figure 4) in each class were included, the ROC AUC was 0.8 and PR AUC was 0.79. The performance of the model was potentially degraded by three possibilities:•That an uncertain region lay between abs∆_UPDRS_ that was a clinically meaningful (Support Class 1) and one which was not clinically meaningful (Support Class 0).•That the MFSL_DT_ did not reflect the subjects’ worse motor function (i.e., they were already “ON” in the sense that their usual level of motor function was not being seen at this time possibly because they had already responded to medications);•Day to day variability in the amplitude MFSL_DT_ and MFSL_ET_, and in latency to peak response, especially from clinical variation in enteric delivery of the drug to absorption sites.

Means for identifying each of these possibilities were developed and then excluded from the comparison to assess their effect on the ROC statistics (Table 8 and Figure 4). In the followings, we elaborate on these exclusion mechanisms.

#### 3.1.1. Subjects in whom there was Uncertainty as to whether abs∆_UPDRS_ was Meaningful

The performance of the model will be enhanced if only subjects whose abs∆_UPDRS_ are clearly positive or clearly negative are included, and cases whose labelling is ambiguous or uncertain are removed. It was noted that (i) the boundary between Level 0 and Level 1 was 10 UPDRS III points (Table 3); (ii) MFSL were separated by 12.5 UPDRS III points; (iii) inspection of Figure 3 suggests that a clinical “uncertain range” also exists (see further discussion in Section 4.2). Thus, a region from 11–14 and centered on 12.5 would remove most “uncertain” cases, and uncertain abs∆_UPDRS_ were defined as being 11 ≤ abs∆_UPDRS_ ≤ 14 [25]. That is, for the purpose of an LDCT, a meaningful improvement was defined as abs∆_UPDRS_ > 14 (Class 1) and an insignificant improvement as abs∆_UPDRS_ ≤ 10 (Class 0). Figure 4A shows that these “uncertain” cases did indeed have an abs∆_PKG_ that was intermediate between cases in Class 0 and 1. When uncertain abs∆_UPDRS_ were removed, the ROC AUC and PR AUC improved respectively, to 0.83 and was 0.81.

#### 3.1.2. Subjects whose Motor Function at dose Time did not Represent their Worst Motor Function Level

If MFSL_DT_ was not the most severe level of motor function present in a subjects PKG, it suggests that they may be “already ON”: that is, they may have already received treatment. This is relevant because the aim clinically, when performing an LDCT, is for motor function at the time of the dose during the LDCT to represent the PwP’s worst motor function (highest score) and to this end, anti-PD medications are not taken after the prior evening. Motor function is usually assessed several hours after awakening so any “so-called” sleep benefit has dispersed [32,33].

When performing an LDCT, the “OFF” state is considered to represent the subject’s worst motor function. Similarly, the motor function at DT during a PKG is assumed as being the time of maximum motor dysfunction. Subjects were excluded if they were known to routinely take medications prior to the first reminder, however on inspection of the PKG of some cases, it was apparent that higher levels of bradykinesia occurred later in the day and so it was possible that medications were consumed prior to the first reminder without this being recorded. This possibility was examined further. First subjects cases whose MFSL_DT_ was in the treated range were found: That is MFSL_DT_ was less than MFSL 3 (Table 3, and see reference [15] for rationale for choosing this level). The next step was to establish whether motor function deteriorated significantly later in the day by estimating the subjects MFSL (as per the model: average plus one standard deviation) from 46 min after the first dose until 18:00 h. If this was more than 1 MFSL greater than MFSL_DT_, they were flagged as “already ON”: 19% of PwP met this criterion and its effect was analyzed using the ROC statistic (Table 8). Exclusion of these “already ON” subjects improved the ROC AUC to 0.87 and PR AUC was 0.85.) When cases that were “already ON” and those that were in the uncertain abs∆_UPDRS_ were both excluded, the ROC AUC and PR AUC further improved to 0.89 and 0.87 respectively.

Sixty-eight percent of all PwP in this study were on D2 agonists and the percent of PwP taking D2 agonists who were flagged as “already ON” was also 32%. Furthermore, the average LED was 177 in those who already ON, compared to 170 in the rest of the cohort. D2 agonist contributed less than 20% of the total levodopa equivalent dose in most PwP (Figure 2, median 15% (8–30% IQR)). Neither abs∆_PKG_ and abs∆_UPDRS_ were smaller when the size or the relative contribution of the D2 agonist dose was large. This suggested that D2 agonists did not contribute to whether a LR could be assessed using the PKG.

#### 3.1.3. Subjects with Excess Variability in the Amplitude and Latency to Peak Response

In terms of applying the model, estimates of MFSL_DT_ or MFSL_ET_ could be affected in some subjects by epochs excluded because of inactivity or exercise. This affected the sample size and increased the variability in estimates of motor function at DT and ET. Also, the size and time to peak response to a dose of levodopa (such as in an LDCT) will be affected by variability in gut motility and gastric emptying causing erratic delivery of levodopa to transport sites in the gut, which would result in day to day variability even if measured by UPDRS III. There is also variation in the UPDRS III itself with variation in assessment by an individual performer and between performers. Excess variability in amplitude of response was assessed by measuring the standard deviation in the MFSL_DT_ and MFSL_EF_ of all epochs in DT and ET of all available days. If both were greater than 1 MFSL (Table 3) then it was deemed that there is significant variability in the motor function at both times: 23% of PwP had excess variability in amplitude. Variability in latency from dose to peak was also assessed: cases with excess variability were cases where the standard deviation of the estimation of MFSL was greater than 1 for both DT and ET. Thirty five percent of PwP were flagged as having increased variability in latency to peak. Exclusion of cases with variability measured in this way, in addition to cases that were “already ON” and were in the uncertain abs∆_UPDRS_ resulted in ROC AUC of 0.92 and PR AUC of 0.87 (Table 8).

### 3.2. Classification Performance of the %∆_PKG_ in Predicting the %∆_UPDRS_ in the LDCT

As discussed above, it is common practice clinically to use the %∆_UPDRS_ as an outcome of the LDCT. While the performance of the model in predicting the percent improvement in motor function was not as effective as predicting the change in motor function (Table 9 and Figure 4), it nevertheless resulted in ROC AUC and PR AUC of 0.82 and 0.73 (respectively) when cases that were “already ON”, in the uncertain zone and with excess variability were excluded.

### 3.3. Clinical Relevance of the Measuring LR with the PKG

The duration of disease was plotted against the LR measured by abs∆_PKG_ and abs∆_UPDRS_ (Figure 5). In this study the abs∆_PKG_ increase with disease duration, which is similar to reports of others using abs∆_UPDRS_ [4,34], whereas the abs∆_UPDRS_ showed only a weak trend to increase with disease and possibly the variation between different clinical reporters of the UPDRS III may have obscured this trend.

The average time to peak response was 45 min (±19 SD) and this compares to 51 min (±25 SD) reported elsewhere [35]. This also provides support that the PKG is indeed measuring an LR similar to that measured by the LDCT.

The size of the first dose used on the PKG ranged widely with a median of 150 mg (interquartile range ±50 mg). The minimum was 50 mg and the 10th percentile was 100 mg and 90th 250 mg. These doses are less than those used by most clinics including the three that provided participants for this study. However, there was a non-significant trend for the median abs∆_PKG_ to be larger when the 1st dose LED was 150 mg or less compared >200 mg. There was a very similar pattern to the median of the abs∆_UPDRS_ and the size of the 1st dose of the morning.

The abs∆_UPDRS_ decreases following DBS due to improved motor function in the “off” state [36]. Thus, it would be expected that abs∆_PKG_ will reduce following DBS. PKGs performed before and 6 months after DBS were available for estimation of their abs∆_PKG_ (Figure 5C,D). As expected, there was marked reduction in abs∆_PKG_ and the size of the reduction was predicted by the pre-surgery LR as measured by the abs∆_PKG_ (Figure 5D).

The data regarding the size of the LR in relationship to age and the response to DBS are presented here to show that the findings of the instrumented assessment of the LR reproduce the findings found by UPDRS III.

## 4. Discussion

The aim of this study was to establish whether data from the PKG could be used to assess LR with a similar classification performance to the LDCT. We designed a Machine Learning model to define 6 levels of motor function severity using UPDRS III scores from before and after a dose of levodopa. The models are designed and validated against unseen data to ensure their generalizability. Employing these models, we could use the PKG data to predict the abs∆_UPDRS_ with ROC AUC of 0.92 that suggests PKG can be used to accurately replicate LDCT in an ambulatory fashion. One of the factors that contributing the difficulty in modelling the LDCT is the wide variety of clinical practice, both in the literature and in the clinics participating in this study. There are differences as to whether the LR is expressed as an absolute difference in the scores (abs∆) [34,37] or as a percentage of the “OFF” score (%∆) [2,38,39]. There is also variation in what constitutes a significant response: a %∆ of 30% is widely accepted [40,41] although changes including 20% [42], 25% [43], 33% [2], >40% [44] and 25–50% [45] are cited. Some centers measure the “ON” state at a specific time [2,38,39], typically 45 min [46,47], whereas others establish a peak score. There is also no uniformity in the size of the dose: most use an absolute dose [40,41,48] ranging from 150–400 mg of levodopa but others use some multiple of the usual morning dose [49,50,51].

### 4.1. Steps That Improve the Classification Performance of the abs∆PKG_PKG_

While the ROC AUC of all subjects was acceptable (0.80), substantial improvement (ROC AUC = 0.92) could be obtained by excluding those in the uncertain abs∆_UPDRS_ range, those who were “already ON” with deterioration observed later in the day and those with excess variability in response. The issue of an “uncertain zone” in the classification of the abs∆_UPDRS_ is addressed in Section 4.2.

#### 4.1.1. Addressing the “Already ON” Issue 

When an LDCT is performed, there is an expectation that “OFF” motor function represents the PwP’s worst motor function. To achieve this, anti-PD medications are not taken after the prior evening and the test is usually performed several hours after awakening so any “so-called” sleep benefit [32,33] has dispersed. Similarly, for the PKG, the intention was that the level of motor function at the DT would represent the worst motor function experienced over the course of the day. In many cases, medications were taken immediately on awakening so, if “sleep benefit” were a real entity, it would interfere with the model’s estimations. We are aware of the discourse relating to this entity [32,33] and merely note here that it is possible that it could have prejudiced the performance of the model. However, prior consumption of medications is a more tangible problem. Subjects were excluded from this study if the clinical notes indicated routine consumption of medications (including apomorphine) prior to the first reminder because this would improve the motor function at DT. However, it is possible that some subjects consumed medications prior to their first reminder without this being recorded.

Whatever the mechanism, subjects whose bradykinesia has been partially alleviated at the time of the first PKG reminder could be revealed as having adequately treated bradykinesia at the time of the first dose but having worse levels of motor function later in the day. The statistical method used in this study identified subjects whose level of motor function was already in the treated range and also experienced higher levels of bradykinesia (as identified by the PKG) later in the day. These subjects were flagged as “already ON” at DT: 19% of PwP met this criterion and excluding these PwP improved the ROC AUC and PR AUC (to 0.87 and 0.85 respectively).

The main purpose of introducing this flag was to indicate that early morning alleviation of bradykinesia through whatever mechanism (e.g., medication or sleep benefit) reduced the performance of detecting a levodopa response with the PKG. In applying the PKG as a test of levodopa responsiveness in the future, this would be addressed by specifically requesting PwP to refrain from medications and to be “up and active” for some time prior to the first reminder. The latter point would not only address the potential contribution of sleep benefit but would also reduce the number of epochs removed because of inactivity or sleep (see Section 3.1.1.). This proposition should be addressed in a future study. However, the flag should remain as a marker to question the compliance of individuals in relation to medications consumption in the morning.

#### 4.1.2. Addressing the Issue of Increased Variability

While the effect of variability in gut motility on response to levodopa is well recognized [52], it is not commented on as a source of error in interpretation of the LDCT. Performing the study first thing in the morning is thought to reduce variability in gastric emptying. Arguably however, the average time and amplitude of response from 6 days may be a truer representation than a single sample as in the LDCT. Another important factor is that many PwP, are often still in bed or at least inactive at the time of the first dose. As a result, many epochs are excluded because inactivity indicates that they cannot be used to assess bradykinesia. The smaller sample size leads to increased variability. In the future this could be simply addressed by requesting that PwP are awake and active for half an hour prior to the first dose, which is much less intrusive than asking them to attend hospital in the untreated state for an LDCT. Removal of cases with excess variability had the least impact on improving the classification performance.

### 4.2. Reporting Absolute Size or Percentage Change of LR

The justification for including an “uncertain zone” was both empirical in that it improved the ROC statistics, but also because there is clinical uncertainty about what constitutes a positive response to levodopa. Clinical convention usually recognizes a significant LR as %∆_UPDRS_ of 30% [40,41] but range from 20–50% have been reported [2,42,43,44,45]. The literature is not forthcoming as to why a percentage improvement is preferred. In clinical practice, LDCT is most commonly performed in subjects with significant levels of bradykinesia in the untreated state because the questions for performing the LDCT relate to suitability for advanced therapy or the diagnosis of responsive parkinsonism. For example, in this study the mean UPDRS_OFF_ was 48 (±13 SD) and less than 10% had UPDRS_OFF_ < 30. The abs∆_UPDRS_ increases with duration of disease, most likely due to increasing severity of UPDRS_OFF_ [34,37]. Pieterman et al. [4] recently reported that abs∆_UPDRS_ correlated better with the UPDRS_OFF_ (Figure 5) and with a range of motor and non-motor scores than did the more commonly used %∆_UPDRS_. While both %∆_UPDRS_ and abs∆_UPDRS_ were compared with the LR measured by the UPDRS III, we also found that the abs∆_UPDRS_ from the PKG provided the best predictor. While PwP in the early stages of disease do have a measurable LR, the abs∆_UPDRS_ is less than 11 UPDRS III points (~7) and the %∆_UPDRS_ cutoff for clinically meaningful results is around 33% [41]. Figure 3 indicates that PwP whose disease duration was less than 5 years were more likely (Fishers, *p* = 0.015) to have scores that were not clinically significant in absolute terms and this would argue that a different model is required for early disease.

The justification for the size of the zone (4 UPDRS III units) and its location on the UPDRS III scale (11–14) was empirical, drawing on the observations described in Section 3.1.1 and led to an “uncertain zone” of 11 ≤ abs∆_UPDR_ ≤ 14 being used. The model would be strengthened if a future study included a greater number of cases with parkinsonism of whom clinicians are confident that the LR is not clinical meaningful: these cases might include multi-systems atrophy, progressive supranuclear palsy or vascular parkinsonism.

The current model suits the clinical question as to whether the LR is suitable for deploying advanced therapies such as deep brain stimulation. In these subjects, fluctuations are well developed and LR is usually large. However, evidence of LR is sometimes sought in early PD for both clinical and research reasons and would argue for a separate model to deal with small amplitude changes in early disease.

### 4.3. Pharmacological Agents Prescribed while Measuring the LR with the PKG

The median size of the first dose used on the PKG was less than those used by most clinics including the 3 that provided participants for this study. However as noted above, there appears to be little uniformity in the size of the dose: The use of the smaller dose may not affect the interpretation of a LDCT [53] and does not appear to have adversely affected the modelling. In further studies it might be advisable to use the PKG to ensure that the first morning dose is successful in alleviating bradykinesia. This would ensure that whatever the size of the 1st morning dose, it is large enough to produce a clinically meaningful improvement.

There is also no clarity around handling of D2 agonists for a LDCT [40,48,49], let alone the PKG’s version of the LDCT. Many D2 agonists have a half-life that is long enough to have lingering effects in the morning if taken the night before. The CAPSIT protocol [2] and others [50] recommend ceasing for 24 h agonists before the LDCT but few clinics adhere to this. They did not appear to affect the outcome when using the PKG data to model the LDCT.

### 4.4. Routine Reporting of Levodopa Responsive with the PKG

While this study was intended to test the feasibility of measuring LR using the PKG system, it is relevant to speculate on how it may be incorporated into routine clinical care. If future studies confirm that the prediction of an individual PwP’s LR was accurate providing the PwP is awake and active from 30 or more minutes prior to consuming the first dose since retiring the previous evening, until the dose is effective (~2 h later), then the LR could be considered as part of routine monitoring. Under these circumstances, it might be possible to report the abs∆_PKG_ and “OFF” and “ON” PKG levels. An indicator of the reliability of the measure based on the presence of excessive inactivity and sleep or the probability that earlier dose has been taken despite advice to the contrary might also be possible. Such a reporting mechanism is subject to further.

## 5. Conclusions

A Machine Learning model using features from the PKG can be used to predict the LR obtained from a LDCT. The performance of the test for individual PwP is likely to be very high if two practical steps were taken:•PwP are awake and active for 30 or more minutes prior to consuming the first dose.•No PD medications are taken between retiring and the first dose.

These suggestions should be confirmed in future studies which also include a greater proportion of people with parkinsonian features that do not respond to levodopa. It does not appear that that a first dose of under 200 mg levodopa substantially alters the modelling, but future studies may be needed to establish a more certain method for ensuring the first dose is adequate for eliciting an LR. This model was built with the assumption that a clinically useful LR was greater than 14 UPDRS III points. The LR in early PD and some other conditions is smaller and a suitable model could be built in the future to address this need.

In summary, used appropriately this approach seems likely to be effective in establishing levodopa responsiveness in most cases providing the PKG is performed appropriately.

## Figures and Tables

**Figure 1 sensors-19-05153-f001:**
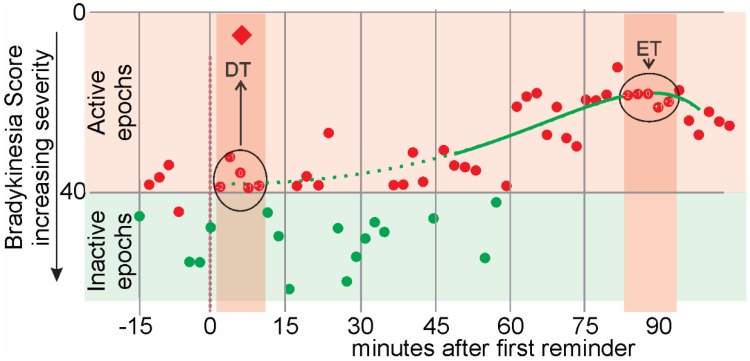
Stylized representation of one day of the Parkinson’s Kinetigraph (PKG) recording. The Y axis shows the PKG’s bradykinesia score in bradykinesia score (BKS) units and the X axis is time in minutes, before after the first reminder of the morning (red vertical dotted line at “0” time). The acknowledgement that the dose was consumed is shown as a red diamond. The dots represent individual BKS for each two-minute epoch: green dots represent epochs that lie in the “inactive” range and red within the active range. The green line represents the smoothed time series from all 6 days of recording, with the heavy line being from 46 min to 90 min after the first acknowledgement of the reminder (red diamond at ~6 mins). The apricot shading area shows the ten minutes (5 epochs, circled) used to establish the bradykinesia scores defined in Section 2.3 at the time of the dose (DT) and around the time of the peak (ET).

**Figure 2 sensors-19-05153-f002:**
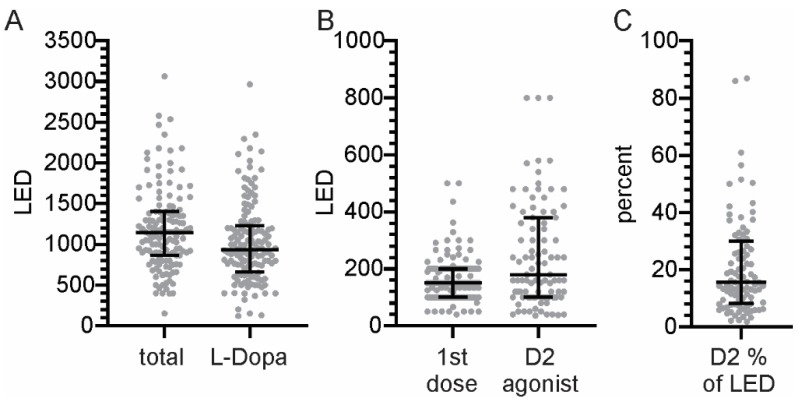
(**A**) shows the range of total Levodopa Equivalent dose (LED) and the dose of levodopa (L-Dopa). (**B**) shows the LED from the first dose of levodopa (1st dose) and from D2 agonists over the course of the day. (**C)** shows the percentage of the LED contributed to by D2 agonists.

**Figure 3 sensors-19-05153-f003:**
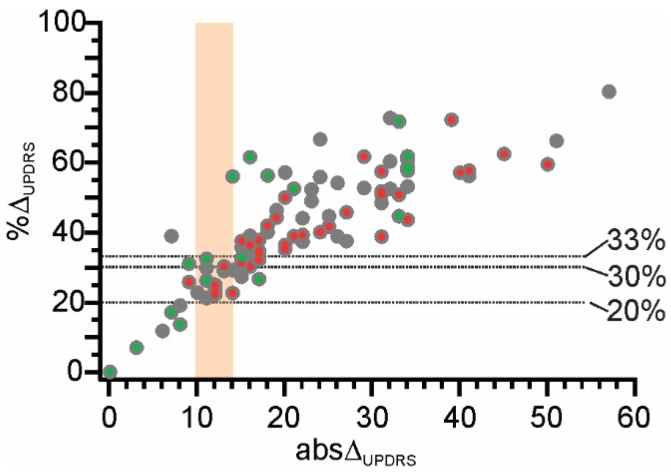
This shows the relationship between %∆UPDRS (Y axis) and abs∆UPDRS (X axis). The grey circles represent all PwP. Gray dots with a green center are subjects with disease duration of 4 years or less and those with a red center are PwP with disease duration of 10 or more years. The vertical apricot shaded region shows the “uncertain” zone (see Section 3.1.2). To the right of this shade area are cases where the abs∆UPDRS was considered to be a clinical meaningful increase (see text for criteria) whereas to the left, abs∆UPDRS was not clinically meaningful. The 3 horizontal lines indicate the three commonly used %∆UPDRS, showing that a region of clinical uncertainty also exists.

**Figure 4 sensors-19-05153-f004:**
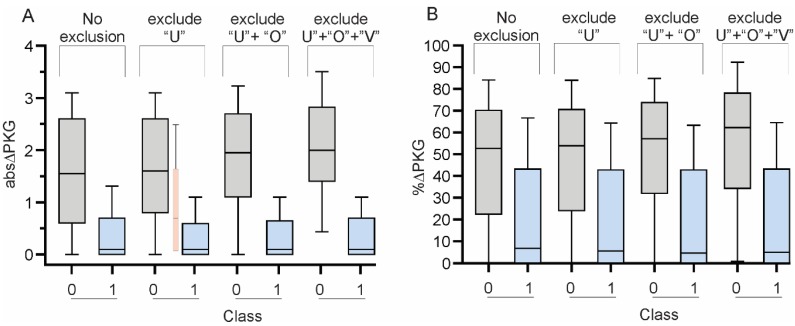
These two plots are box and whiskers plots of the distribution of Class 0 and Class 1 in Table 8 (**A**) and 10 (**B**) plotted according the 4 corresponding column groups in that table: “U” indicates “uncertain”, “O” indicates “Already ON” and “V” indicates “Variable”. The small, pink shaded “box and whiskers” plot, between Case 0 and Case 1 in (**A**) in the Exclude U group shows the distribution of abs∆PKG of the uncertain cases. The boxes are the median and quartiles with the “whiskers” showing the 90th and 10th percentile.

**Figure 5 sensors-19-05153-f005:**
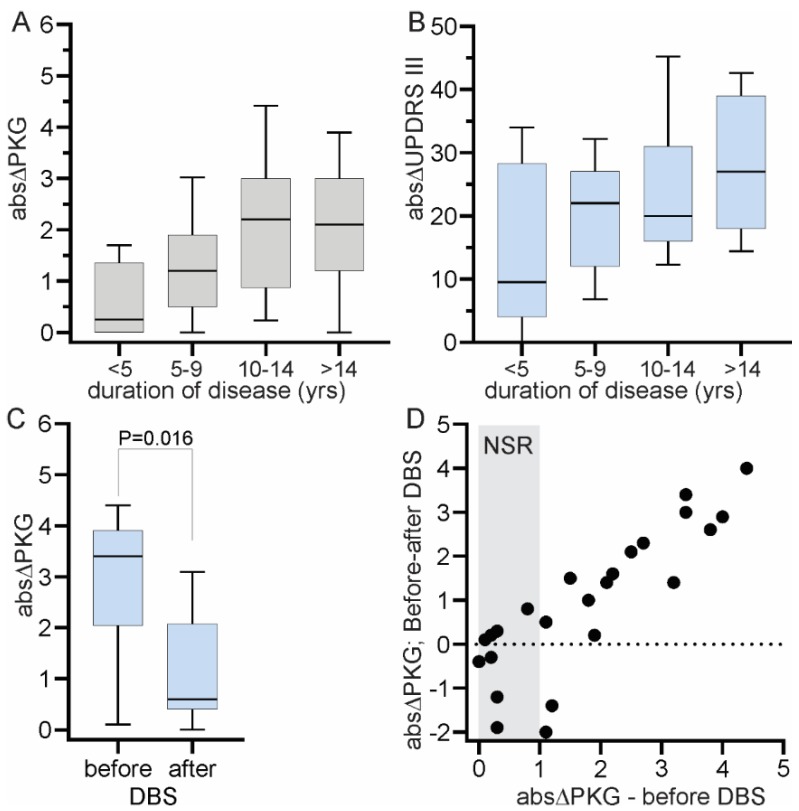
(**A**,**B**) show the change in LR according to duration of disease (in years). (**A**) shows the abs∆_PKG_ and 5B shows the abs∆_UPDRS_. (note that 1 unit on the Y axis of (**A**) approximates 12 UPDRS III units shown on the Y axis of (**B**)). (**C**) shows abs∆_PKG_ before and after deep brain stimulation (DBS). (**D**) shows the same data, with the difference in abs∆_PKG_ before and after DBS (X axis) plotted against the abs∆_PKG_ before DBS. The grey region marked NSD (no significant response) in (**D**) indicates the absolute delta before DBS that is not significant. In (**A**–**C**) the boxes are the median and quartiles with the “whiskers” showing the 90th and 10th percentile.

**Table 1 sensors-19-05153-t001:** Clinical features of Person with Parkinson’s Disease (PwP) having levodopa challenge test (LDCT).

Clinical Parameter	Clinic (Mean ± SD (Number of Case Where Data is Available)		
	1	2	3
UPDRS III ^ϕ^ OFF	54 ± 15 (53)	51 ± 11 (49)	41 ± 12 (49)
UPDRS III ^ϕ^ ON	27 ± 10 (53)	34 ± 9 (49)	18 ± 8 (49)
abs∆_UPDRS_ ^χ^	27 ± 11	17 ± 10	24 ± 8
% ∆_UPDRS_ *	51 ± 14	32 ± 15	58 ± 13
Age (years)	61.6 ± 6.7 (53)	65.6 ± 9.1 (49)	58.8 ± 12.0 (2)
Disease duration (years	10.0 ± 4.1 (47)	9.9 ± 5.1 (49)	10.8 ± 6.0 (49)
D2 agonist LED ^§^	20 ± 16 (53)	10 ± 12 (49)	16 ± 32 (49)
LED	869 ± 386 (53)	1226 ± 745 (49)	979 ± 544 (49)
LED 1st dose	138 ± 55 (50)	194 ± 114 (49)	162 ± 56 (49)

^ϕ^ Unified Parkinson’s Disease Rating Scale Part III. ^§^ expressed as a percentage of total the total levodopa equivalent dose (LED) in mg is calculated according to Tomlinson et.al. [12]. ^χ^ absolute difference between “OFF” to “ON” UPDRS III. * difference between “OFF” to “ON” UPDRS III expressed as percent of “OFF” UPDRS III.

**Table 2 sensors-19-05153-t002:** Clinical features of PwP having LDCT.

Criteria	No of Subjects Meeting Criteria				
	Clinic 1	Clinic 2	Clinic 3	Total PwP	Controls
basic criteria	53	49	49	151	174
Plus ^§^ Dose time motor function (DT)	47	47	47	141	132
Plus Effect Time motor function (ET)	51	48	49	148	170
Plus DT & ET (for LDCT)	46	47	47	140	46

^§^ “Plus” indicates that this line includes subjects who met basic criteria PLUS the added criteria in the column. DT=Dose Time which is the 5 epochs (10 min), centered on the acknowledgement to the first reminder of the day and shown as a red diamond in Figure 1. It may be a few minutes after the reminder was delivered (as in Figure 1). ET=Effect Time, which is the time when levodopa had its peak effect. It is calculated for the PKG as the peak of Smoothed Weekly BKS Time series from 46 min to 90 min after the acknowledgement of the dose reminder.

**Table 3 sensors-19-05153-t003:** Support sets for the 6 target classes.

LEVEL	0	1	2	3	4	5
Number or Class Labels	305	54	80	76	46	29
UPDRS III Interval	0–10	10–22.5	22.5–35	35–47.5	47.5–60	≥60

**Table 4 sensors-19-05153-t004:** Candidate predictor features and their MI of candidate features with six target classes.

Feature Name	Description	MI
*BK*	*BK Score*	*0.04*
*BKS_M10P*	*30 min window moving 10th percentile of BKS*	*0.3*
*BKS_M25P*	*30 min window moving 25th percentile of BKS*	*0.24*
*BKS_M50P*	*30 min window moving 50th percentile of BKS*	*0.17*
BKS_M75P	30 min window moving 75th percentile of BKS	0.16
*BKS_M90P*	*30 min window moving 90th percentile of BKS*	*0.22*
BKS_WM10P	30 min window weighted moving 10th percentile of BKS	0.22
BKS_WM25P	30 min window weighted moving 25th percentile of BKS	0.20
BKS_WM50P	30 min window weighted moving 50th percentile of BKS	0.17
*BKS_WM75P*	*30 min window weighted moving 75th percentile of BKS*	*0.18*
BKS_WM90P	30 min window weighted moving 90th percentile of BKS	0.2
TA	Tremor Amplitude	0.04
TA_M50P	30 min window moving 50th percentile of Tremor Amplitude	0.07
TA_WM50P	30 min window weighted moving 50th percentile of Tremor Amplitude	0.08
*TA_WM50P_Log*	*Log10(1 + 30 min window weighted moving 50th percentile of Tremor Amplitude)*	*0.1*

Rows in italics were features selected in the refinement process.

**Table 5 sensors-19-05153-t005:** Joint Mutual Information Ranking of refined candidate features and the MI of their Weekly Aggregate Mean with six target classes.

Feature	Joint MI Ranking	MI (Weekly Aggregate)
*BKS_M10P*	*1 (0.3)*	*0.17*
*BKS_M25P*	*2 (0.3)*	*0.19*
BKS_M90P	3 (0.28)	0.04
BKS_M50P	4 (0.24)	0.12
BKS_WM75P	5 (0.19)	0.06
*TA_WM50P_Log*	*6 (0.06)*	*0.09*

Rows in italics were features selected following assessment of relevancy and redundancy.

**Table 6 sensors-19-05153-t006:** Cross validation (CV) and training performance metrics on Weekly Aggregate Mean predictions for candidate classifier models.

			Classifier				
			1	2	3	4	5
Logistic Regression	ROC AUC ^§^	CV	0.77	0.79	0.87	0.87	0.83
		Train	0.78	0.8	0.87	0.88	0.85
	PR AUC ^Φ^	CV	0.75	0.78	0.81	0.72	0.58
		Train	0.76	0.79	0.83	0.74	0.59
PCA + Logistic Regression	ROC AUC	CV	0.77	0.79	0.87	0.87	0.83
		Train	0.77	0.79	0.87	0.88	0.85
	PR AUC	CV	0.75	0.77	0.81	0.72	0.58
		Train	0.75	0.78	0.83	0.74	0.6
SVC-RBF Kernel	ROC AUC	CV	0.69	0.77	0.85	0.84	0.66
		Train	0.76	0.78	0.87	0.87	0.91
	PR AUC	CV	0.67	0.75	0.78	0.69	0.54
		Train	0.74	0.75	0.81	0.76	0.75
Gradient Boosting Decision Trees	ROC AUC	CV	0.77	0.77	0.86	0.86	0.8
		Train	0.8	0.81	0.89	0.89	0.88
	PR AUC	CV	0.75	0.75	0.79	0.69	0.57
		Train	0.79	0.8	0.85	0.78	0.65

^§^ ROC AUC: area under curve of Receiver Operating Characteristics. ^Φ^ PR AUC: Area under the Precession Recall curve.

**Table 7 sensors-19-05153-t007:** The comparison of test and train performance illustrates an overall low variance and anticipates high generalizability of the performance for any unseen data. The ROC AUC or PR AUC performance on train set is occasionally better than on test set as they vary for different models leading to difficult or outlier examples randomly fall in either sets (see Section 2.8).

		Classifier				
		1	2	3	4	5
ROC AUC	Train Set	0.78	0.8	0.87	0.88	0.85
	Test Set	0.79	0.88	0.85	0.83	0.82
PR AUC	Train Set	0.76	0.79	0.83	0.74	0.59
	Test Set	0.79	0.88	0.83	0.7	0.65

**Table 8 sensors-19-05153-t008:** Classification performance, in terms of ROC and Precision-Recall statistics, of the PKG predictions of the size of the UPDRS III response in the LDCT.

Metric	All	Excluded Cases			
		Already ”ON” ^§^	Uncertain ^ϕ^	Already ”ON” ^§^ & Uncertain ^ϕ^	Already ”ON” ^§^ & Uncertain ^ϕ^ & Variable ^χ^
ROC AUC	0.8	0.87	0.83	0.89	0.92
PR AUC	0.79	0.85	0.81	0.87	0.87
Support Class 0 (total n)	198	193	188	185	181
PwP (n)	24	19	14	11	7
Controls (n)	174	174	174	174	174
Support Class 1 (n)	116	95	107	88	50

^§^ Subjects whose motor function at Dose Time did not represent their worst motor function level. ^ϕ^ Subjects where there was uncertainty as to whether ∆UPDRSIII was meaningful. ^χ^ Day to day variability in the amplitude and latency to peak response to levodopa.

**Table 9 sensors-19-05153-t009:** Classification performance, in terms of ROC and Precision-Recall statistics, of the PKG predictions of the percentage improvement of the UPDRS III response in the LDCT.

Metric	All	Excluded Cases			
		Already ”ON” ^§^	Uncertain ^ϕ^	Already ”ON” ^§^ & Uncertain ^ϕ^	Already ”ON” ^§^ & Uncertain ^ϕ^ &Variable ^χ^
ROC AUC	0.74	0.78	0.76	0.79	0.82
PR AUC	0.71	0.74	0.72	0.75	0.73
Support Class 0 (total)	205	198	192	186	181
Support Class 0 (PwP)	31	24	18	12	7
Support Class 0 (controls)	174	174	174	174	174
Support Class 1 (PwP)	109	90	103	87	50

^§^ Subjects whose motor function at Dose Time did not represent their worst motor function level. ^ϕ^ Subjects where there was uncertainty as to whether ∆UPDRSIII was meaningful. ^χ^ Day to day variability in the amplitude and latency to peak response to levodopa.

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
