# Peer review of "Prediction of the Levodopa Challenge Test in Parkinson’s Disease Using Data from a Wrist-Worn Sensor"

_sensors, 2019, doi:10.3390/s19235153_

Round 1
Reviewer 1 Report
This paper addresses the issue of predicting the result of the Levodopa Challenge Test in Parkinson's Disease using data acquired with a Parkinson's Kinetigraph.
The overall quality of the paper is fairly good; authors describe their work, with convincting arguments in a structured way.
I have some minor comments :
There are erroneous references in the text (e.g. on lines 335, 370, 374) There is also a missing reference to the table 7 on line 398 In Table 1, units are missing The last paragraph before subsection 2.5 (lines 241-246) is not perfectly clear. I believe that it could be improved by explaining the protocol followed in each clinic separately. In Table 2, it should be mentionned in the caption that numbers given correspond to the number of subjects In subsection 2.6, a classification is given with 6 levels (0 to 5) depending on value of UPDRS III score. There is no clinical evidence on how the scale was designed, how thresholds were defined. Does it make sense to have the 0 category defined for UPDRS III<10 ? Doest it make sense to have the same difference between all categories. PR AUC is used to evaluate the performance of the models, but F Measure, which is a very common measure to balance recall and precision, is not provided. If possible, authours should ask the F Measure for every test. Authors never explain that logistic regresssion has been used in the 2.10 subsection. Even if it is a consequence of 2.9 subsection, it should be clearly written. It would be of great interest of adding a subsection in the discussion about Human Factors and Usability issues of the system. Authors have conclusions on the limits of the system, but there is no feedback on the feasibility of their recommendations on the conditions to use their system. There are some spelling/english errors: a closing bracket is missing on line 224 There is a wrong closing bracket on line 271 line 357 : A train set and a test set was were defined
Regarding these comments, the paper should be ready for publication.
Author Response
We thank Reviewer 1 for their thoughtful consideration of our manuscript. Our response to their questions and comments are indicated below.
There are erroneous references in the text (e.g. on lines 335, 370, 374)
Response: – this change has been made along with merging of Sections 2.8, 2.9 and 2.10 into Section 2.9
There is also a missing reference to the table 7 on line 398
Response: d – this change has been made along with merging of Sections 2.8, 2.9 and 2.10 into Section 2.9
In Table 1, units are missing UPDRS has no units.
Response:
Years has been added to age and duration. Mg has been added at the explanation of LED. The last paragraph before subsection 2.5 (lines 241-246) is not perfectly clear. I believe that it could be improved by explaining the protocol followed in each clinic separately.
Response: Changed to Line 244
2.5. The LDCT.
As described earlier, the LDCT is performed by administering a dose of levodopa in the “OFF” state which is then compared with the “ON” state using the UPDRS III. As already noted, there is considerable variation in the performance of this test, which was also apparent in the procedures used by the three clinics participating in this study.
OFF state assessment. All clinics assessed the “OFF” state at least 12 hours after the PwP’s last PD medication. No clinic ceased D2 agonist for more than 12 hours. Loading dose. One clinic gave a supramaximal dose of 50% more than their usual first dose of levodopa. One clinic gave 20% more than the usual morning dose in dispersible form. The third reviewed the “ON” state on their usual dose ON state assessment. Two clinics assessed the “ON” state following the levodopa dose: one from 45 mins on and the other at about 50-60 minutes. The third clinic made the assessment on another day to when the “OFF” state was assessed.
In Table 2, it should be mentionned in the caption that numbers given correspond to the number of subjects
Response: Table 2 has been changed to reflect this.
In subsection 2.6, a classification is given with 6 levels (0 to 5) depending on value of UPDRS III score. There is no clinical evidence on how the scale was designed, how thresholds were defined. Does it make sense to have the 0 category defined for UPDRS III<10 ? Doest it make sense to have the same difference between all categories.
Response: We have expanded the existing discussion relating to the thresholds for the various UPDRS III severity categories. Unfortunately, there is very little clinical information to guide us in this choice. We have re-written lines 358-364 to better reflect the clinical basis for this choice
Unfortunately, the literature is relatively silent on the UPDRS III of normal subjects, and while a score of UPDRS III<10 for MFSLUPDRS “0” is somewhat arbitrary, it is designed to be conservatively low, based on the range of scores in newly diagnosed subjects[1] and on the clinical experience of the authors. The size of the increments in UPDRS III between higher MFSLUPDR relate to the upper limits of inter/intra rater variability[2, 3] and clinical important differences[4], although we accept that there is little literature to guide this choice.
PR AUC is used to evaluate the performance of the models, but F Measure, which is a very common measure to balance recall and precision, is not provided. If possible, authours should ask the F Measure for every test.
Response: We agree with the reviewers comment that f1 score is a balanced performance measure for imbalanced label distributions and provides a balanced measure of precision and recall (harmonic mean of precision and recall) and to avoid true negatives positively biasing. However, f1 score is calculated per probability cut-off threshold, which will be highly affected by the application, context of use and dataset distribution to balance between false positives and false negatives, and true positives the performance (as in ROC AUC). We chose to report average measures to ensure covering the use of different probability thresholds in different applications. Therefore, in order to address the balance between precision and recall, we use the average precision at different recalls also averaged on the two classes which will even further address this.
We refer the reviewer to the following paragraph in Section 2.9: Line 446-451
“We report two performance measures: (i) area under curve of Receiver Operating Characteristics (ROC AUC) which provides an average Sensitivity at different Specificities; (ii) area under curve of Precision (ratio of true positives to all positive declarations including false positives) vs. Recall (=Sensitivity) (PR AUC) which provides an average Precision at different Recalls averaged over the two classes. The performance metric (ii) is reported to account for the class imbalance that varies over the five Classifiers”
Authors never explain that logistic regresssion has been used in the 2.10 subsection. Even if it is a consequence of 2.9 subsection, it should be clearly written.
Response: Section 2.8, 9 & 10 have been merged as Section 9 and Lines 461-465 now states:
The Logistic Regression model performs as well, if not better than the other models and is simpler in that it enhances the likelihood of generalisability and interpretability of the algorithm. Thus, the Logistic Regression model was further analysed for validation and bias-variance assessment using the five classifier models and Logistic Regression was used in the following sections of this study.
It would be of great interest of adding a subsection in the discussion about Human Factors and Usability issues of the system. Authors have conclusions on the limits of the system, but there is no feedback on the feasibility of their recommendations on the conditions to use their system.
Response: The following para has been inserted at Line 795
4.4 Routine reporting of Levodopa responsive with the PKG.
While this study was intended to test the feasibility of measuring LR using the PKG system, it is relevant to speculate on how it may be incorporated into routine clinical care. If future studies confirm that the prediction of an individual PwP’s LR was accurate providing the PwP is awake and active from 30 or more minutes prior to consuming the first dose since retiring the previous evening, until the dose is effective (~2 hours later), then the LR could be considered as part of routine monitoring. Under these circumstances, it might be possible to report the absDPKG and “OFF” and “ON” PKG levels. An indicator of the reliability of the measure based on the presence of excessive inactivity and sleep or the probability that earlier dose has been taken despite advice to the contrary might also be possible. Such a reporting mechanism is subject to further.
There are some spelling/english errors: a closing bracket is missing on line 224 (Now 275)
Response - bracket inserted
There is a wrong closing bracket on line 271 (now line 343)
Response – changed to opening bracket
line 357 : A train set and a test set was were defined (now line 440)
Response – changed to “A train set and a test set were defined”
Simuni, T., A. Siderowf, S. Lasch, C. S. Coffey, C. Caspell-Garcia, D. Jennings, C. M. Tanner, J. Q. Trojanowski, L. M. Shaw, J. Seibyl, N. Schuff, A. Singleton, K. Kieburtz, A. W. Toga, B. Mollenhauer, D. Galasko, L. M. Chahine, D. Weintraub, T. Foroud, D. Tosun, K. Poston, V. Arnedo, M. Frasier, T. Sherer, S. Chowdhury, K. Marek, and Initiative Parkinson's Progression Marker. "Longitudinal Change of Clinical and Biological Measures in Early Parkinson's Disease: Parkinson's Progression Markers Initiative Cohort." Mov Disord 33, no. 5 (2018): 771-82. Goetz, C. G., G. T. Stebbins, and B. C. Tilley. "Calibration of Unified Parkinson's Disease Rating Scale Scores to Movement Disorder Society-Unified Parkinson's Disease Rating Scale Scores." Mov Disord 27, no. 10 (2012): 1239-42. Post, B., M. P. Merkus, R. M. de Bie, R. J. de Haan, and J. D. Speelman. "Unified Parkinson's Disease Rating Scale Motor Examination: Are Ratings of Nurses, Residents in Neurology, and Movement Disorders Specialists Interchangeable?" Mov Disord 20, no. 12 (2005): 1577-84. Shulman, L. M., A. L. Gruber-Baldini, K. E. Anderson, P. S. Fishman, S. G. Reich, and W. J. Weiner. "The Clinically Important Difference on the Unified Parkinson's Disease Rating Scale." Arch Neurol 67, no. 1 (2010): 64-70.

Reviewer 2 Report
This study is interesting, well-driven, with a high number of subjects included.
My major remark concerns the length of the paper, that contains a lot of information and results. I feel that it could be an improvement to shorten irrelevant or redundant parts. I propose below some ideas, but authors can propose other ones.
My other remarks and questions are given below.
Introduction:
L52-77: This part may be reduced, or some details could be moved further.
L78-85: On the contrary, part devoted to systems of the literature is too short. Other studies dealing with motion analysis with PD should be cited.
L82: I propose to write "The PKG is obtained by a Person With Parkinson (PwP) disease...
L92-100: This paragraph gives the results of the study, and is not suitable at this place. It also gives the names of the three clinics, that are also given in lines 103-104. Instead of it, you should give a description of the organization of your paper.
Section 2
L141: It could be interesting to place an image of the PKG system.
L193: Sentence is not correct and I did not see a green arrow in Figure 1.
L335, L370, L374: link error.
L346: I wonder if it is necessary to talk about PCA that does not improve the results (surprisingly). To simplify, and since no justification about that is given, this information may be resumed in only one sentence.
L379-383: Section 2.9 is short. Could you merge it with previous or following on?
L387-397: This paragraph is globally a repetition of things already written before.
L398: The Table number is missing.
L399: They are no comments on Table 7. Some values are greater in Test set than in Train test.
Section 3
L413: They --> The
Results and Discussion
L607: Sentence "Preforming... morning" might not be correct.
Author Response
We thank Reviewer 2 for their thoughtful consideration of our manuscript. Our response to their questions and comments are indicated below.
The length of the paper.
Response: We have made many changes including combining sections and deleting redundant sections.
Introduction:
L68-77: This part may be reduced, or some details could be moved further.
Response: Line 58-65. This has been substantial reduced although some of the information has been inserted in the Discussion (Lines 595-603).
L78-85: On the contrary, part devoted to systems of the literature is too short. Other studies dealing with motion analysis with PD should be cited.
Response: For two reasons, we respectfully disagree with the reviewer that we should systematically review other motion analysis studies of PD. First, this is not a review of the various means of measuring PD with sensors and such a discussion would substantially lengthen the Introduction but not be relevant to the question of whether an LDCT could be modelled if such a system was available. We have in fact cited 3 reviews of motion systems for PD. The second reason is that while there are several such systems, to our knowledge, none, other than the PKG system measure bradykinesia at home while the subject is performing routine activities of daily living. Instead we have inserted a brief section, that is more explicit on why we have focussed on the PKG. Line 66-73 now reads:
The recent development of objective measures of the motor features of PD raises the possibility of assessing LR in the home were reviewed in [1-3]. However, these reviews indicate, there are only one or two of these systems that actually measure bradykinesia while subjects are engaged in routine daily activities. One of these ambulatory measuring systems is the Parkinson's Kinetigraph (PKG), which scores the motor features of PD[4, 5], without the need for the subject to engage in specific test activities. The data available from a PKG recording includes the response to the first dose of levodopa each day for 6 consecutive days.
L82: I propose to write "The PKG is obtained by a Person With Parkinson (PwP) disease...
Response –Line 72: changes are made as suggested.
L92-100: This paragraph gives the results of the study, and is not suitable at this place. It also gives the names of the three clinics, that are also given in lines 103-104. Instead of it, you should give a description of the organization of your paper.
Response –Line 83-86 have been changed as follows.
In this paper, we describe how data from the UPDRS III in the “OFF” and “ON” state (UPDRSIIION and UPDRSIIIOFF) during an LDCT and features from the PKG data were used to build a model of motor function severity levels (MFSL) using Logistic Regression. This model was then used to predict absD and %D from the LDCT calculated using the PKG data.
Section 2
L141: It could be interesting to place an image of the PKG system.
Response: The logger is only one component of the system and looks like a watch. (http://www.globalkineticscorporation.com.au/). Rather than display a commercial product we suggest a reference to the companies web site. The reader was already directed to the website at the bottom of the paragraph (Line 214).
Line 194 has been changed to say
The logger is a smart watch that is worn on the most affected wrist (see http://www.globalkineticscorporation.com.au/ for details).
L193: Sentence is not correct and I did not see a green arrow in Figure 1.
Response: Line 241-244 have been changed (below)- there is no green arrow in Figure 1
Reminder Time. The PKG logger is programmed to vibrate at specific times to remind subjects to take their medications. Subjects can acknowldge when they actually consume the medications by “swipping “ the smart screen on the watch. The first acknowledgement of the reminder is shown as a red diamond in Figure 1.
L335, L370, L374: link error.
Corrected
L346: I wonder if it is necessary to talk about PCA that does not improve the results (surprisingly). To simplify, and since no justification about that is given, this information may be resumed in only one sentence.
We agree with the reviewer’s comment that the use of PCA is extensively discussed while it is not applied in the final results. However, other models such as Gradient Boosting Trees and Support Vector Classifier are also not used in the final model. We have only incorporated the Logistic Regression + PCA model to address the fact that Logistic Regression assumes that the features should not have collinearity. The rationale to uphold this assumption is that the occurrence of multi-collinearity will eventually degrade the model performance and generalizability. We ensure that resolving this issue through the use of PCA does not improve performance and generalizability. Therefore, we conclude that multi-collinearity that exists in our feature set does not practically affect generalizability of the model while we can still use the two features which were intentionally designed to capture the distribution of BK score while they have structural collinearity. To address reviewer’s concern, we added the following sentence to the manuscript to Section 2.9 (lines 453-457):
First it is evident that reducing the dimensionality of the structurally collinear features with PCA did not improve the performance of the Logistic Regression. This will support the employment of BKS_M10P and BKS_M25P together as separate features in Logistic Regression model that allows to capture the distribution of BK score intuitively reflected in these two features.
L379-383: Section 2.9 is short. Could you merge it with previous or following on?
Sections 2.8, 2.9 and 2.10 have all been merged as Section 2.9?
L387-397: This paragraph is globally a repetition of things already written before.
Response: removed as part of the merger above.
L398: The Table number is missing.
Response: Fixed
L399: They are no comments on Table 7. Some values are greater in Test set than in Train test.
Response: As for each classifier (1 to 5) train and test sets are randomly chosen to have a balanced distribution of class 0 and class 1, the train and test sets are different for each classifier. Therefore, more difficult examples or outliers can interchangeably switch between train and test set for different classifiers leading to the classifier performing worse on train set on some cases. The balanced distribution of these difficult or outlier examples was not possible before we perform a full analysis of the levodopa response. So, we had to do the random selection of train and test set blind to these considerations.
To address reviewer’s concern, we added more description to Table 7
Table 7 The comparison of test and train performance illustrates an overall low variance and anticipates high generalisability of the performance for any unseen data. The ROC AUC or PR AUC performance on train set is occasionally better than on test set as they vary for different models leading to difficult or outlier examples randomly fall in either sets (see Section 2.8).
We also modified line 441-443 in Section 2.9:
For each binary classification problem, we randomly allocate 30% samples from each class around Dose and Effect Time of the entire set to test set and use the remaining in train set. As binary classes for each classifier are different, train and test set will randomly vary across them.
Section 3
L413: They --> The
Corrected (now line 498)
Results and Discussion
L607: Sentence "Preforming... morning" might not be correct.
Corrected (now line 727)
Maetzler, W., J. Domingos, K. Srulijes, J. J. Ferreira, and B. R. Bloem. "Quantitative Wearable Sensors for Objective Assessment of Parkinson's Disease." Mov Disord 28, no. 12 (2013): 1628-37. Maetzler, W., J. Klucken, and M. Horne. "A Clinical View on the Development of Technology-Based Tools in Managing Parkinson's Disease." Mov Disord 31, no. 9 (2016): 1263-71. Espay, A. J., P. Bonato, F. B. Nahab, W. Maetzler, J. M. Dean, J. Klucken, B. M. Eskofier, A. Merola, F. Horak, A. E. Lang, R. Reilmann, J. Giuffrida, A. Nieuwboer, M. Horne, M. A. Little, I. Litvan, T. Simuni, E. R. Dorsey, M. A. Burack, K. Kubota, A. Kamondi, C. Godinho, J. F. Daneault, G. Mitsi, L. Krinke, J. M. Hausdorff, B. R. Bloem, and S. Papapetropoulos. "Technology in Parkinson's Disease: Challenges and Opportunities." Mov Disord (2016). Griffiths, R. I., K. Kotschet, S. Arfon, Z. M. Xu, W. Johnson, J. Drago, A. Evans, P. Kempster, S. Raghav, and M. K. Horne. "Automated Assessment of Bradykinesia and Dyskinesia in Parkinson's Disease." J Parkinsons Dis 2, no. 1 (2012): 47-55. Farzanehfar, P., and M. Horne. "Evaluation of the Parkinson's Kinetigraph in Monitoring and Managing Parkinson's Disease." Expert Rev Med Devices 14, no. 8 (2017): 583-91.
